# Loss of *Ecrg4* improves calcium oxalate nephropathy

**Daniela Cabuzu**[1,2], **Suresh K. Ramakrishnan**[1,2], **Matthias B. Moor**[1,2¤a], **Dusan Harmacek**[1,2], **Muriel Auberson**[1], **Fanny Durussel**[1,2], **Olivier Bonny**[1,2,3¤b]*

1 Department of Biomedical Sciences, University of Lausanne, Lausanne, Switzerland, 2 The National Centre of Competence in Research (NCCR) "Kidney.CH - Kidney Control of Homeostasis", Zurich, Switzerland, 3 Service of Nephrology, Department of Medicine, Lausanne University Hospital, Lausanne, Switzerland

¤a Current address: Department of Nephrology and Hypertension, Bern University Hospital, Bern, Switzerland
¤b Current address: Service of Nephrology, Department of Medicine, Fribourg State Hospital, Fribourg, Switzerland
* Olivier.Bonny@unil.ch

**Data Availability Statement:** All relevant data are within the article and its Supporting information files.

**Funding:** This work was supported by the Swiss National Science Foundation, research grant

## Abstract

Kidney stone is one of the most frequent urinary tract diseases, affecting 10% of the population and displaying a high recurrence rate. Kidney stones are the result of salt supersaturation, including calcium and oxalate. We have previously identified Esophageal cancer-related gene 4 (*Ecrg4*) as being modulated by hypercalciuria. *Ecrg4* was initially described as a tumor suppressor gene in the esophagus. Lately, it was shown to be involved as well in apoptosis, cell senescence, cell migration, inflammation and cell responsiveness to chemotherapy. To the best of our knowledge, nothing is known about ECRG4's function in the renal tissue and its relationship with calciuria. We hypothesized that the increased expression of *Ecrg4* mRNA is triggered by hypercalciuria and might modulate intratubular calcium-oxalate precipitation. In this study, we have first (i) validated the increased *Ecrg4* mRNA in several types of hypercalciuric mouse models, then (ii) described the *Ecrg4* mRNA expression along the nephron and (iii) assessed ECRG4's putative role in calcium oxalate nephropathy. For this, *Ecrg4* KO mice were challenged with a kidney stone-inducing diet, rich in calcium and oxalate precursor. Taken together, our study demonstrates that *Ecrg4*'s expression is restricted mainly to the distal part of the nephron and that the *Ecrg4* KO mice develop less signs of tubular obstruction and less calcium-oxalate deposits. This promotes *Ecrg4* as a modulator of renal crystallization and may open the way to new therapeutic possibilities against calcium oxalate nephropathy.

## Introduction

Prevalence of chronic kidney diseases and of kidney stones is constantly increasing [1]. In both conditions, supersaturation of calcium-oxalate salts and crystal formation may result

310030-182312 and by the special program NCCR Kidney.CH to OB.

either in obstruction of renal tubules or in stone formation. Thus, hypercalciuria and hyperoxaluria are two major risk factors for crystallogenesis in urine [2].

We have previously identified the *Ecrg4* gene as upregulated in hypercalciuric mice. Highly conserved from fish to humans, *Ecrg4* encodes a 148 amino acids precursor protein that can be processed into various peptides, without identified regulatory proteases [3]. *Ecrg4* mRNA can be found in various tissues, such as brain, esophagus, heart and kidney, whereas, the protein ECRG4 was documented in the heart and adrenal gland [4]. The functions of ECRG4 and derived peptides are still largely unknown. It was first speculated that *Ecrg4* might be a tumor suppressor gene, due to its decreased expression in esophageal cell carcinomas, while still detectable in the normal esophageal epithelia surrounding the tumor [5]. Yet, *in vivo* studies have extended the anti-tumoral role of *Ecrg4* to glioma and to prostate, colorectal and breast cancer [6–8]. Further, *Ecrg4*'s overexpression was shown to affect apoptosis, cell senescence, cell migration, inflammation and cell responsiveness to chemotherapy [3]. Its activity depends on its cellular localization, secretion and processing, which argues for a non-traditional tumor suppressor action, and rather for a role close to a cytokine/chemokine. Downstream genes that were shown to be activated by *Ecrg4* include *Nfkb1*, *Nfkb2*, *P53* and *Cox2* [3].

*Ecrg4*'s role in the kidney and especially in hypercalciuria is unknown. Due to its higher expression in a hypercalciuric mouse model, we hypothesized that *Ecrg4* mRNA is triggered by hypercalciuria and might modulate kidney stone formation. In the present work, we first describe *Ecrg4* mRNA expression in the kidney. Second, we show that *Ecrg4* expression is increased in various hypercalciuric mouse models. Third, we assessed the potential role of *Ecrg4* in kidney stone formation by challenging *Ecrg4* KO and control mice with a diet inducing hypercalciuria and hyperoxaluria. We found substantial protection against calcium oxalate nephropathy in male mice devoid of *Ecrg4*. In total, this piece of data points towards *Ecrg4* as a potential regulator of calciuria and stone formation.

## Materials and methods

### Animals

The State veterinarian Office (Office vétérinaire cantonal, Canton de Vaud, Switzerland) approved all animal studies (authorization numbers VD3261 and VD2987). All breeding and cohort colonies were hosted in our animal facility under approved protocols. Mice were housed four to five per cage, with free access to water and food (standard mouse diet #3800, KLIBA, Kaiseraugst, Switzerland), in a temperature and humidity-controlled room with an automatic 12/12-h light/dark cycle. Heterozygous *Ecrg4* mice were purchased from the Mutant Mouse Regional Resource Center (B6;129S5-1500015O10Rik[tm1Lex]/Mmucd). In this model, coding exon 1 of the RIKEN cDNA 1500015O10 gene was targeted by homologous recombination. Mice were backcrossed for 10 generations and heterozygous males and females were bred in order to obtain *Ecrg4*[-/-] (KO) mice. Littermates *Ecrg4*[+/+] have been used as controls for all experiments. Kidney-specific sodium-calcium exchanger 1 (NCX1) *Ncx1*-knockout mice (Pax8-LC1-Cre NCX1fl/fl) were also used in the study (unpublished, manuscript in preparation). Claudin-2 (*Cldn2*) KO mice (B6; 129S5-Cldn2[tm1Lex]/Mmucd) were purchased from Mutant Mouse Regional Resource center.

### Microdissection of nephron segments

Mice were deeply anesthetized by intraperitoneal injection with Ketanarkon (Streuli Pharma AG, Uznach, Switzerland; 100 µg/g body weight) and Rompun (Bayer, Leverkusen, Germany; 10 µg/g body weight), and the left kidney was perfused first with PBS, then with DMEM/F12 (1:1, Life Technologies, Carlsbad, USA) supplemented with 40 mg/ml of Liberase Blendzyme 2

(Roche, Basel, Switzerland). Thin slices of the kidney were cut along the corticomedullary axis and incubated 40–60 min at 37°C in the same DMEM/F12/Liberase medium. Then, the kidney slices were transferred into DMEM/F12 (1:1) alone to stop the digestion, and the microdissection was performed under a magnifier in ice-cold 0.05%BSA/DMEM/F12. The proximal convoluted tubule (PCT), the thick ascending limb (TAL), distal convoluted tubule and cortical connecting tubule (DCT-CNT), and the cortical collecting duct (CD) were isolated based on their specific morphology.

## RNA extraction and quantitative PCR

**Microdissected tubules.**   Immediately after microdissection, tubules were subjected to RNA extraction using the RNeasy Micro Kit (# 74004, Qiagen, Hilden, Germany) according to the manufacturer instructions.

**Other tissues.**   RNA from kidneys of mice injected with $1,25(OH)_2$-vitamin D and PTH as previously described [9], was used to assess the Ecrg4 expression. Tissues were homogenized in TRIzol$^®$ Reagent solution (#15596026 Thermo Fisher, Waltham, USA) followed by extraction with 1-bromo-3-chloropropane reagent (BCP, Molecular Research Center, Cincinnati, USA) and isopropanol precipitation. RNA (1 μg) was reverse transcribed using PrimeScript RT reagent kit (#RR037B Takara Biotechnology, Otsu, Japan) according to manufacturer's guidelines. TaqMan Gene Expression Assays (#4370074, Applied Biosystems, Warrington,UK) was used to detect *Ecrg4* (Mm00470447_m1), together with *Actb* (Mm4351315_m1). Other genes were quantified using SYBR green PCR master Mix (#4367659, Applied Biosystems, Warrington,UK). The primers used are listed in Table 1. Primers were ordered from Microsynth (Balgach, Switzerland). Quantitative real-time PCRs were carried out on an ABI PRISM 7500 equipment (Applied Biosystems, Warrington,UK). If the cycle threshold (CT) were >36, the expression was considered as null. The relative expression of the genes was calculated using the comparative $2^{-\Delta\Delta CT}$ method.

## Metabolic and endocrine studies

Male and female mice aged 6 weeks were housed individually in metabolic cages (Tecniplast, Buguggiate, Italy) for 2 days in order to be habituated before baseline urine collection, with free access to water and food. After the habituation period, control and *Ecrg4* KO mice were fed either the chow diet or the crystal forming diet CaOx (1.5% Calcium Chloride and 2% Hydroxyl-L-proline) for 8 days. Twenty-four hour urine collection was repeated on day 1 and day 8, by housing the mice in metabolic cages.

Urine sodium and potassium concentrations were determined by flame photometry (Instrumentation laboratory), while the urine creatinine, calcium and magnesium concentrations were measured in the Laboratoire Central de Chimie Clinique, Lausanne University Hospital (Lausanne, Switzerland). Urine osmolality was measured with an automated freezing point osmometer (Advanced model 2020 Multi-Sample Osmometer, Advanced Instruments, Norwood, USA) and the urine pH using a pH-meter (#6.0224.100, Metrohm, Herisau, Switzerland). The Laboratoire Central de Chimie Clinique, Lausanne University Hospital (Lausanne, Switzerland), measured all the blood parameters unless otherwise specified. Creatinine concentration was measured using the enzymatic method.

For the dihydrotachysterol experiment, the chow diet of control male mice was supplemented with 1.5 mg/kg dihydrotachysterol (D9257, Merk, Kenilworth, USA) during 7 consecutive days.

Treatments with 1,25(OH)2-vitamin D or PTH were previously reported in [9]. In brief, male C57BL/6N mice aged 13 to 15 weeks received a subcutaneous injection of 2 μg/kg body

**Table 1. Primers sequences.**

| | | | |
|---|---|---|---|
| *Atp2b4* F | CTT AAT GGA CCT GCG AAA GC | *Atp2b4* R | ATC TGC AGG GTT CCC AGA TA |
| *Aqp2* F | TTC GAG CTG CCT TCT ACG TG | *Aqp2* R | GGA AGA GCT CCA CAG TCA CC |
| *Actb* F | GTC CAC CTT CCA GCA GAT GT | *Actb* R | AGT CCG CCT AGA AGC ACT TGC |
| *Cyp24a1* F | GAA GAT GTG AGG AAT ATG CCC TAT TT | *Cyp24a1* R | CCG AGT TGT GAA TGG CAC ACT |
| *Cyp27b1* F | ATG TTT GCC TTT GCC CAG AG | *Cyp27b1* R | GAC GGC ATA TCC TCC TCA GG |
| *Calb1* F | ATT TCG ACG CTG ACG GAA GT | *Calb1* R | GTG GGT AAG ACG TGA GCC A |
| *Cldn2* F | AAG GTG CTG CTG AGG GTA GA | *Cldn2* R | AGT GGC AGA GAT GGG ATT TG |
| *Cldn14* F | ACC CTG CTC TGC TTA TCC | *Cldn14* R | GCA CGG TTG TCC TTG TAG |
| *Cldn16* F | CAAACGCTTTTGATGGGATTC | *Cldn16* R | TTTGTGGGTCATCAGGTAGG |
| *Casr* F | CAC AGT TGC CTT GTG ATC CTC | *Casr* R | ATG CAG AGG TGT AGG GTG GT |
| *Cox2* F | GCC TAC TAC AAG TGT TTC TTT TTG CA | *Cox2* R | CAT TTT GTT TGA TTG TTC ACA CCA T |
| *Nfkb1* F | GAA ATT CCT GAT CCA GAC AAA AAC | *Nfkb1* R | ATC ACT TCA ATG GCC TCT GTG TAG |
| *Nfkb2* F | CTG GTG GAC ACA TAC AGG AAG AC | *Nfkb2* R | ATA GGC ACT GTC TTC TTT CAC CTC |
| *Pvalb* F | CAG CGC TGA GGA CAT CAA GA | *Pvalb* R | AGT CAG CGC CAC TTA GCT TT |
| *Pthr1* F | CAG ACG ATG TCT TTA CCA AAG | *Pthr1* R | TCC ACC CTT TGT CTG ACT CC |
| *Slc9a3* F | CTT CAA ATG GCA CCA CGT CC | *Slc9a3* R | AAT AGG GGG CAG CAG GTA GA |
| *Slc8a1* F | AGA GCT CGA ATT CCA GAA CGA TG | *Slc8a1* R | TTG GTT CCT CAA GCA CAA GGG AG |
| *Trpv5* F | TCC TTT GTC CAG GAC TAC ATC CCT | *Trpv5* R | TCA AAT GTC CCA GGG TGT TTC G |
| *Vdr* F | GGA TCT GTG GAG TGT GTG GAG ACC | *Vdr* R | CTT CAT CAT GCC AAT GTC CAC GCA G |

weight 1,25(OH)$_2$-vitamin D$_3$ (#D1530, Sigma Aldrich, St. Louis, USA), dissolved in ethanol 1% and 99% of NaCl 0.9%. Control mice were injected with 1% (v/v) ethanol in NaCl 0.9%. The mice were dissected 6 hr after injection.

For the analysis of organs from PTH-treated animals, material harvested from a previous study was used [9]. In this study, male 12–13 weeks old C57BL/6N mice received a subcutaneous injection of 80 μg/kg body weight human PTH 1–34 (#P3796, Sigma Aldrich, St. Louis, USA) dissolved in NaCl 0.9% or NaCl 0.9% alone as vehicle. Animals were dissected 2 hours after injection.

For the acute furosemide experiment, male C57BL/6N mice aged 12 weeks received a single intraperitoneal injection (Lasix, Sanofi-Aventis, Switzerland; 20mg/kg body weight) diluted in saline solution. The control group received an intraperitoneal injection of NaCl 0.9%. The mice were sacrificed after 15min, 30min, 60min or 240min and the kidneys were immediately harvested.

For the chronic furosemide experiment, male C57BL/6N mice aged 12 weeks received a daily intraperitoneal injection for 6 days of either furosemide diluted in saline solution (20mg/ kg body weight), or saline solution for the control mice. The mice were sacrificed at day 7 and the kidneys were harvested.

## Kidney histology and crystals

Left kidneys were fixed in 4% paraformaldehyde, embedded in paraffin and 5μm sections were cut. Sections were then stained with Pizzolato and H&E. Images were taken on Leica DMi8 microscope. To determine the area of calcium oxalate crystals (stained in black by Pizzolato) over the kidney's surface area, images were obtained at x10, and analyzed with ImageJ. The result was expressed as percentage of black area over the whole kidney area. Similarly, ImageJ software was used for the analysis of tubular dilatation on H&E-stained kidney sections. The investigator was blinded of the code when analysis were performed.

## Western blot and immunostaining

Antibodies against ECRG4 used in the study were from the following sources: Santa Cruz Biotechnology (H-118, USA), Sigma Aldrich (HPA008546, St. Louis, USA), LS Bio (LS-C172856, Seattle, USA), Phoenix Pharmaceuticals (012–25, Mannheim, Germany) and actin (#A2066, Sigma Aldrich, St. Louis, USA). For each of the antibodies, we have used dilutions in the range of 1:500 to 1:50. Each of the primary antibody was used for Western blot, immunohistochemistry and immunofluorescence. Secondary antibodies used were Jackson Immuno Research Anti/Rabbit (111-035-003) or Anti-Mouse (115-035-003), AlexaFluor488 (1:2000 diluted in blocking buffer without BSA, Invitrogen, Carlsbad, USA).

Antibodies against PTHR1 (#sc-12722, Santa Cruz Biotechnology, USA), TRPV5 (developed in house and validated in PMID: 24557712), CLDN2 (#32–5600, Thermo Scientific, Rockford, USA), PVALB, CASR (#MA1-934, Thermo Scientific, Rockford, USA) were used as 1:500 dilution.

For protein extraction, tissues were homogenized in RIPA buffer (Tris pH 7.2 50 mM, NaCl 150 mM, NP40 1%, SDS 0.1%, Na-deoxycholate 0.5% with proteases inhibitor), using TissueLyser (Qiagen, Hilden, Germany). Following the centrifugation (12,000 rpm for 15 min) and supernatants collection, total protein concentration was determined using a BCA protein assay kit (#23227, Thermo Fisher, Waltham, USA). Protein separation was done on 10% and 13% respectively, SDS polyacrylamide gels and blotted onto nitrocellulose membrane (Whatman, Dassel, Germany). Detection was done by chemiluminescence (#34579, Super Signal West Pico, Thermo Scientific, Rockford, USA).

## Immunohistochemistry

Organs were fixed in 4% paraformaldehyde in Phosphate Buffer Solution and then embedded in paraffin before being cut as 5μm-thick sections. Sections were first hydrated gradually through decreasing concentrations of ethanol and then washed in deionized water. Antigen unmasking was done in 10mM sodium citrate buffer, pH 6.0, for 1h at 96˚C, followed by incubation in 0.3% hydrogen peroxide for 10 min at room temperature, and by 1h in blocking buffer (NP-40 0.5%, BSA 2%, normal goat serum (NGS) 3% in PBS). Sections were incubated with primary antibodies for 1h at room temperature. Primary antibodies were washed in PBS three times for 5 min, followed by incubation with anti-mouse or anti-rabbit serum as appropriate for 30 min and developed in 3,3'-diaminobenzidine (DAB substrate kit, #ab64238, Abcam, Cambridge, UK). Lastly, they were counterstained with Nuclear Fast Red Solution (6409-77-4, Merck, Kenilworth, USA).

## Immunofluorescence

Deeply anesthetized mice were perfused via cardiac puncture with 4% paraformaldehyde in PBS. Harvested organs were incubated in 30% sucrose in PBS for at least 24 h before being embedded in Tissue-Tek OCT compound (#4583,Sakura Finetek, Alphen aan den Rijn, The Netherlands) and cut as 7μm-thick sections. Sections were incubated 1 h with blocking buffer (NP-40 0.5%, BSA 2%, normal goat serum (NGS) 3% in PBS) at room temperature, followed by primary antibody incubation overnight at 4˚C. After washing three times with PBS, sections were incubated 1 h at room temperature with secondary antibodies and then washed four times with PBS. The sections were then mounted using Fluoromount-G mounting medium (#0100–01, Southern Biotech, Birmingham, USA). Fluorescent images were visualized using a laser scanning confocal microscope (SP5 AOBS Confocal Microscope, Leica Microsystems, Wetzlar, Germany).

## mCCD cell culture

Mouse cortical collecting duct cells (mCCD) [10] were grown in 6-well plates (#CLS3335, Corning, New York, USA) in growth medium DMEM/F12 1:1 Glutamax (#10565018, Gibco, Waltham, USA) supplemented with insulin (5 ug/ml), human apotransferrin (5 ug/ml), Na-selenate (60 nM), EGF (10 ng/ml), triiodothyronine (1 nM), dexamethasone (50 nM), 1% penicillin/strepromycin, and 2% FCS. Cells were grown to 100% confluence; medium was then exchanged for growth medium containing calcium oxalate monohydrate crystals (#C0350000, Merk, Kenilworth, USA) in a final concentration of 300 or 600 ug/ml; the same volume of PBS 1x was used as a control. Cells were treated over 16 hours, then washed three times with PBS 1x and RNA was isolated using TRIzol®.

## Statistical analyses

Comparisons between groups were conducted using GraphPad Prism software (version 9.0.0). Results are presented, as means ± SD, unless otherwise stated. Statistical analyses were performed using Student's t tests (2-tailed), one-way ANOVA, or two-way ANOVA, using Tukey correction for multiple comparison. Values of $p < 0.05$ were considered statistically significant.

# Results

## *Ecrg4* is expressed in the kidney and along the distal part of the nephron

The expression of *Ecrg4* mRNA in male kidney was analyzed and compared to other organs, including adrenal glands, brain, heart and esophagus. *Ecrg4* mRNA was detected in all tissues tested (Fig 1A). Female kidney and heart tissues have an increased *Ecrg4* expression compared to male mice (S1 Fig). Substantial efforts have been done to identify ECRG4 protein in the same tissues by Western blot or immunostaining. However, none of the antibodies showed specificity in the tested mouse tissues when compared to the same tissues obtained from *Ecrg4* KO mice (S2–S7 Figs).

After confirming the presence of *Ecrg4* mRNA in extracts obtained from the whole kidney, we looked whether *Ecrg4* expression was restricted to certain segmental parts of the nephron or if it was ubiquitous. We microdissected the tubular segments from the kidney of wildtype mice and found that *Ecrg4*'s presence was limited to the thick ascending limb (TAL), distal convoluted tubule/connecting tubule (DCT/CNT) and cortical collecting duct (CD) segments, while no expression was found in the proximal (PROX) tubule (Fig 1B).

## *Ecrg4* transcript is upregulated in hypercalciuric mouse models

We identified *Ecrg4* as an upregulated gene in a RNA-sequencing screen of kidneys from mice devoid of the sodium-calcium exchanger NCX1 and which are hypercalciuric. We wondered whether *Ecrg4* mRNA levels would be increased in other models of hypercalciuria. We analyzed *Ecrg4*'s expression, compared to control littermates in kidney from *Ncx1*-kidney-specific KO mice, Claudin-2 (*Cldn2*) KO mice and mice fed the vitamin D analog, dihydrotachysterol (DHTS). *Ncx1*-kidney-specific KO mice and *Cldn2* KO mice are genetic mouse models of hypercalciuria due to a primary defect in renal tubular (respectively DCT and proximal tubule) calcium transport [11]. DHTS, the vitamin D analogue, is a pharmacological model due to increased intestinal absorption, plasma $Ca^{2+}$ levels and urinary $Ca^{2+}$ excretion [12]. *Ecrg4* mRNA expression was substantially increased in these three chronic hypercalciuric mouse models, as depicted in Fig 2A. We also tested the effect of furosemide—which induces hypercalciuria by blocking calcium reabsorption in the thick ascending limb of Henle- for six

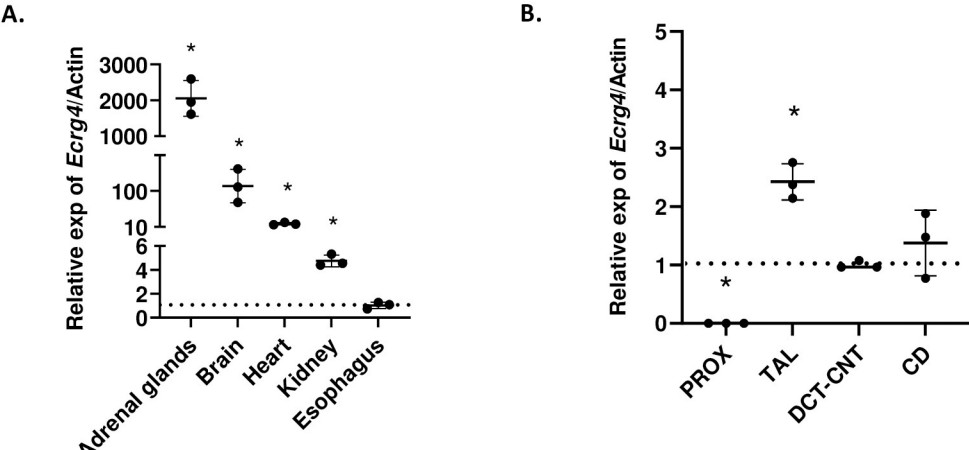

**Fig 1. *Ecrg4* is expressed in the kidney and along the distal part of the nephron. (A)** *Ecrg4* mRNA expression in the indicated organs relative to *Actb* and to the expression in the esophagus (dotted line) in control male mice (n = 3). **(B)** *Ecrg4* mRNA expression in microdissected renal tubular segments related to *Actb* and to the expression in the DCT-CNT segment (dotted line) in control male mice (n = 3). Data are shown as individual dots and mean ± SD. Stars indicate *p<0.05, calculated using t-test. PROX: proximal tubule; TAL: thick ascending limb; DCT-CNT: distal convoluted tubule and connecting tubule; CD: collecting duct.

consecutive days. No effect of furosemide on *Ecrg4* mRNA level was observed. Considering a possible tubular adaptation to hypercalciuria in the case of chronic furosemide treatment, we checked for renal *Ecrg4* mRNA expression after single furosemide treatment. Again, we did not find any change in *Ecrg4*'s expression after acute furosemide, compared to vehicle, up to 240 minutes after furosemide injection (Fig 2B). Hypercalciuria was however substantial only after 15min following the furosemide administration (S8 Fig).

Next, we checked if renal *Ecrg4*'s expression was influenced by the calcitropic hormones 1,25(OH)$_2$-vitamin D and PTH under acute stimulating conditions. No effect of these hormones on the renal *Ecrg4* mRNA level was observed (Fig 2C).

After having observed a response of *Ecrg4*'s expression to chronically induced hypercalciuria, we wondered if the increase in gene expression was segment-specific or was concerning all tubular segment in which *Ecrg4* was found to be expressed. We microdissected the renal tubules of the hypercalciuric kidney-specific *Ncx1* KO mice and found that *Ecrg4* is significantly increased in the TAL, DCT-CNT and CD but not in the proximal tubule compared to control littermates (Fig 2D).

In total, we found that *Ecrg4* expression is increased in the kidneys of several chronic hypercalciuric mouse models and concerns all segments in which *Ecrg4* was found to be expressed, namely TAL, DCT and CD.

## *Ecrg4* KO male mice, but not females, are resistant to induced hypercalciuria

In order to get more insights into the role of *Ecrg4* in calcium handling by the kidney, we obtained *Ecrg4* KO mice and backcrossed them in the C57BL/6 background (Fig 3A). No obvious phenotype (fertility, weight, size, behavior, macroscopic appearance and fur) was observed in KO mice compared to littermate controls. Similarly, no difference in the tested blood and urine parameters was observed (Tables 2 and 3).

To further test the hypothesis that *Ecrg4* modulates calcium-oxalate crystal nephropathy, we fed the mice a diet enriched in calcium (1.5%) and in the oxalate precursor, hydroxyproline

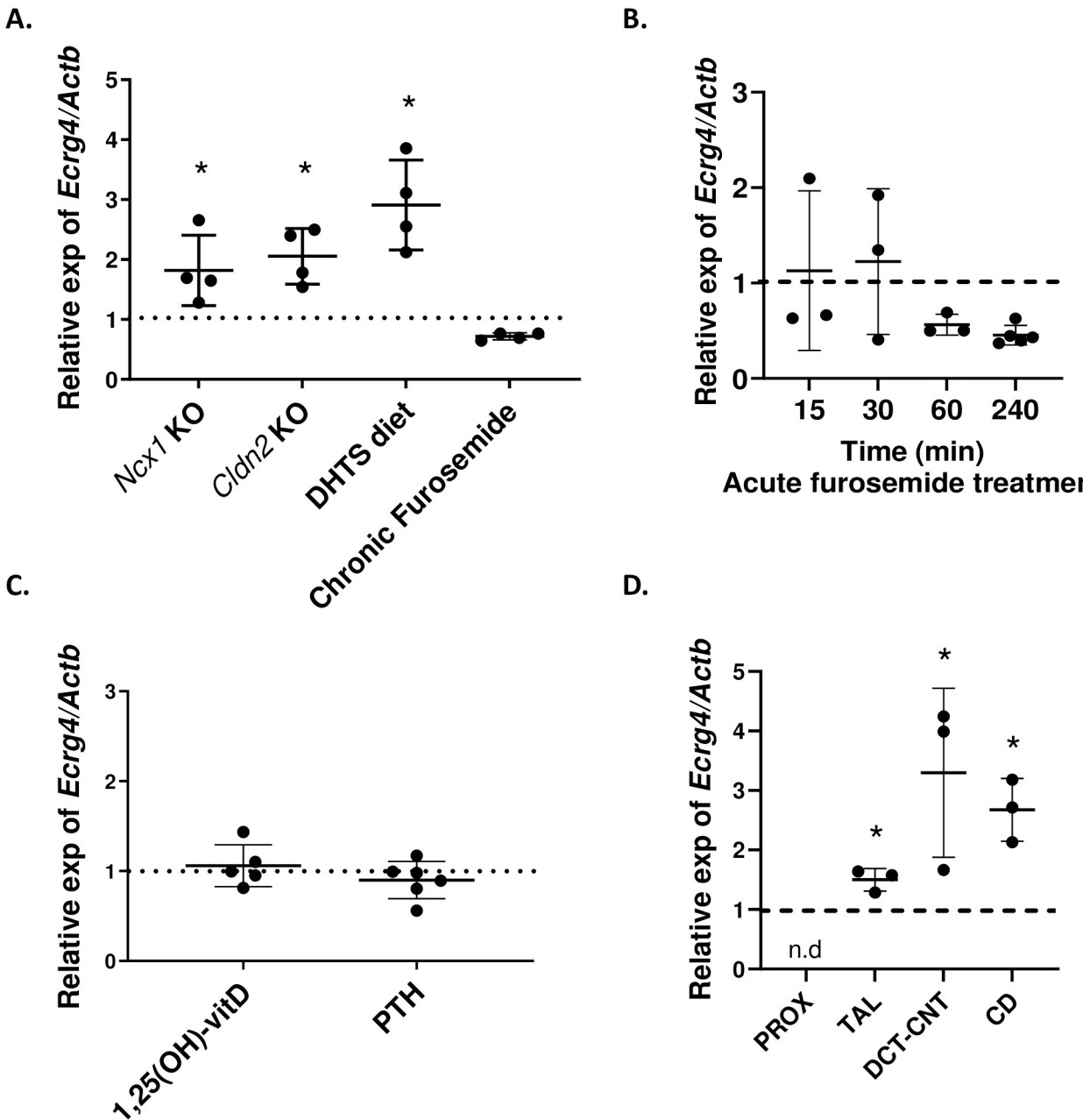

**Fig 2. *Ecrg4* transcript is upregulated in several hypercalciuric mouse models. (A)** *Ecrg4* mRNA expression (related to *Actb* and appropriate control mice for each condition, dotted line) was found increased in the kidneys of mice issued from several established hypercalciuric models (n = 4 males/condition), but not in mice exposed to chronic furosemide (injected with 20mg furosemide/kg body weight for 6 consecutive days). *Ncx1* KO: *Ncx1* KO mice. *Cldn2* KO: Claudin-2 KO mice. DHTS: Dihydrotachysterol-treated mice (1,5mg/kg food) for 7 days. **(B)** Renal *Ecrg4* expression after acute furosemide injection relative to renal expression of vehicle-injected male mice (dotted line). The mice were injected with 20mg furosemide/kg body weight and kidneys were harvested after 15, 30, 60 or 240min (n = 3–5 animals/group). **(C)** Renal *Ecrg4* mRNA expression is not changed by the calcitropic hormones PTH or 1,25(OH)$_2$-vitamin D. The kidneys were harvested 6h after calcitriol injection (n = 5), and 2h after PTH injection (n = 6). The dotted line at 1 represents renal *Ecrg4* mRNA expression of vehicle-treated male mice. **(D)** *Ecrg4*'s regulation along the nephron of the hypercalciuric *Ncx1* KO mouse model compared to control littermates (dotted line) (n = 3 males/condition). Data are shown as individual dots and mean ± SD. Stars indicate *p<0.05, calculated using t test and Dunnett's multiple correction.

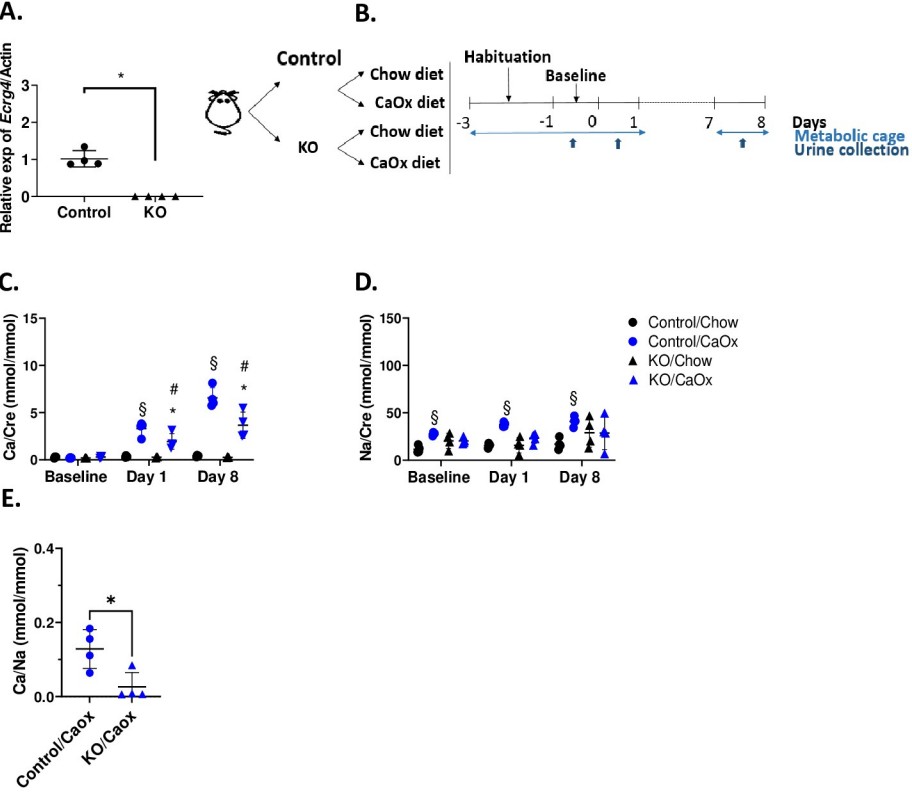

**Fig 3. *Ecrg4* KO male mice are resistant to induced hypercalciuria.** (**A**) *Ecrg4* mRNA expression in the kidney of *Ecrg4* KO mice. P value was calculated using t-test. (**B**) Schematic representation of the experimental plan. Male mice were habituated in metabolic cages for 2 days. Then, control and *Ecrg4* KO mice were exposed to either the regular chow diet or to a diet enriched in calcium (1.5%) and the oxalate precursor hydroxyproline (2%) (CaOx diet) for 8 days. The mice were housed individually in metabolic cages for urine collection at baseline, day 1 and day 8. (**C**) Increase in the urinary calcium-creatinine ratio in both control and *Ecrg4* KO mice under CaOx diet. The ratios were calculated from the urine collected at baseline, day 1 and 8 under chow diet or CaOx diet. At 8 days of the CaOx diet, the *Ecrg4* KO male mice had a significantly lower calcium-creatinine excretion compared to control mice. (**D**) Urinary sodium-creatinine ratio in both control and *Ecrg4* KO male mice. (**E**) Urinary calcium/sodium ratio after 8 days of CaOx diet. Data are shown as individual points and mean ± SD (n = 4). For data B-D, p value was calculated using two-way ANOVA with Tukey correction for multiple comparisons. Stars indicate * $p < 0.05$ between control and *Ecrg4* KO mice; § indicates $p < 0.05$ for comparison of control mice in the two different diets (chow vs. CaOx); # indicates $p < 0.05$ of *Ecrg4* KO mice in the two different diets (chow vs. CaOx).

(2%), further referred to as the CaOx diet (Fig 3B). *Ecrg4* KO male mice fed the CaOx diet had a significantly lower fractional excretion of calcium compared to control mice fed the same diet, despite comparable calcemia and blood creatinine values (Table 2). Such finding was not observed in females (Table 3). Under the CaOx diet, urinary calcium/creatinine ratio was increasing between baseline and day 8 (Fig 3C). Interestingly, after 8 days of treatment, urinary calcium excretion of *Ecrg4* KO male mice was significantly lower compared to control male mice fed the same diet (Fig 3C). As differences of sodium excretion was observed between the groups under CaOx diet (Fig 3D), the urinary calcium/sodium ratio was calculated (Fig 3E) and showed that the lower calciuria observed in *Ecrg4* KO mice is independent of sodium. Female data are provided in supplemental material, but did not show difference in renal calcium handling between *Ecrg4* KO and controls (S9–S11 Figs). Altogether, this data shows that *Ecrg4* KO male mice have lower fractional excretion and lower calciuria than controls while exposed to the CaOx diet.

**Table 2. Plasma chemistry of control and *Ecrg4* KO MALE mice after 8 days of CaOx diet or chow diet.**

| Plasma | Chow diet | | CaOx diet | |
|---|---|---|---|---|
| | Control | *Ecrg4* KO | Control | *Ecrg4* KO |
| Ca$^{2+}$, mM | 2,19±0,06 (4) | 2,21±0,01 (4) | 2,51±0,01 (4) § | 2,42±0,08 (4) |
| Na$^+$, mM | 147,00±2,04 (4) | 150,00±0,70 (4) | 155,75±0,47 (4) § | 155,00±0,70 (4) # |
| K$^+$, mM | 5,00±0,22 (4) | 5,07±0,15 (4) | 4,55±0,09 (4) | 5,12±0,25 (4) |
| PO$_4^{3-}$, mM | 3,25±0,21 (4) | 3,02±0,14 (4) | 1,51±0,15 (4) § | 2,30±0,31 (4) |
| Creatinine, μM | 3,75±0,36 (4) | 5,15±0,95 (4) | 9,50±0,64 (4) § | 8,87±1,68 (4) |
| Ca FE, % | 0,07±0,01 (4) | 0,05±0,00 (4) | 2,48±0,23 (4) § | 1,33±0,31 (4) *# |
| Pi FE, % | 2,16±0,36 (4) | 3.84±0,53 (4) | 3,09±1.09 (4) | 2,61±2,08 (3) |

Data are means ± SEM (n). P values are calculated using one-way ANOVA test, with Tukey correction for multiple comparison.

* indicates $p < 0.05$ between control and *Ecrg4* KO mice;

§ indicates $p < 0.05$ between the two diets for control mice;

# indicates $p < 0.05$ between the two diets for *Ecrg4* KO mice.

## *Ecrg4* KO male mice, but not females, have less renal tubular obstruction under CaOx diet

Control male mice exposed to the CaOx crystal forming diet exhibited intraluminal crystal deposits, clearly detected by Pizzolato staining (Fig 4A) already after 8 days of treatment. The crystals were localized predominantly in the papilla, with fewer crystals observed in the kidney cortex or medulla.

In line with the observed decrease in calcium excretion, *Ecrg4* KO male mice developed less intratubular deposits compared to control mice, with an important inter-individual variability (Fig 4C). Moreover, the tubules were significantly less dilated in *Ecrg4* KO mice compared to control animals (Fig 4B and 4D). Of note, and as often observed in the field, females were completely resistant to crystal formation (S12 Fig). Overall, male mice devoid of *Ecrg4* had less crystal deposits and less tubular dilation, indicative of less tubular plugging.

In order to get more insights into calcium handling by the kidneys of *Ecrg4* KO mice, mRNA expression levels of genes involved in transepithelial calcium transport or its regulation were assessed. No change in *Slc9A3*, *Pthr1*, *Cldn14*, *Cldn2*, *Cldn16*, *Atp2b4*, *Trpv5*, *Calb1*,

**Table 3. Plasma chemistry of control and *Ecrg4* KO FEMALE mice after 8 days of CaOx diet or chow diet.**

| Plasma | Chow diet | | CaOx diet | |
|---|---|---|---|---|
| | Control | *Ecrg4* KO | Control | *Ecrg4* KO |
| Ca$^{2+}$, mM | 2,28 ± 0,02 (5) | 2,13 ± 0,03 (4) | 2,49 ± 0,04 (4) | 2,36 ± 0,06 (7) # |
| Na$^+$, mM | 149,00 ± 0,44 (5) | 150,00 ± 0,49 (5) | 151,00 ± 0,70 (4) | 151,20 ± 0,28 (5) |
| K$^+$, mM | 4,40± 0,19 (5) | 4,00 ± 0,31 (5) | 4,47 ± 0,29 (4) | 4,08 ± 0,23 (5) |
| PO$_4^{3-}$, mM | 2,76 ± 0,13 (5) | 3,02 ± 0,09 (5) | 2,72 ± 0,04 (4) | 2,70 ± 0,17 (7) |
| Creatinine, μM | 5,20 ± 0,58 (5) | 6,40 ± 0,40 (5) | 6,25 ± 0,47 (4) | 4,70 ± 0,60 (4) |
| Ca FE, % | 0,09 ± 0,01 (5) | 0,14 ± 0,01 (4) | 2,45 ± 0,31 (4) § | 2,50 ± 0,23 (4) # |
| Pi FE, % | 6,72 ± 2,36 (5) | 4,83 ± 0,54 (5) | 0,21 ± 0,02 (4) § | 0,16 ± 0,02 (4) |

Data are means ± SEM (n). P values are calculated using one-way ANOVA test, with Tukey correction for multiple comparison.

* indicates $p < 0.05$ between control and *Ecrg4* KO mice;

§ indicates $p < 0.05$ between the two diets for control mice;

# indicates $p < 0.05$ between the two diets for *Ecrg4* KO mice.

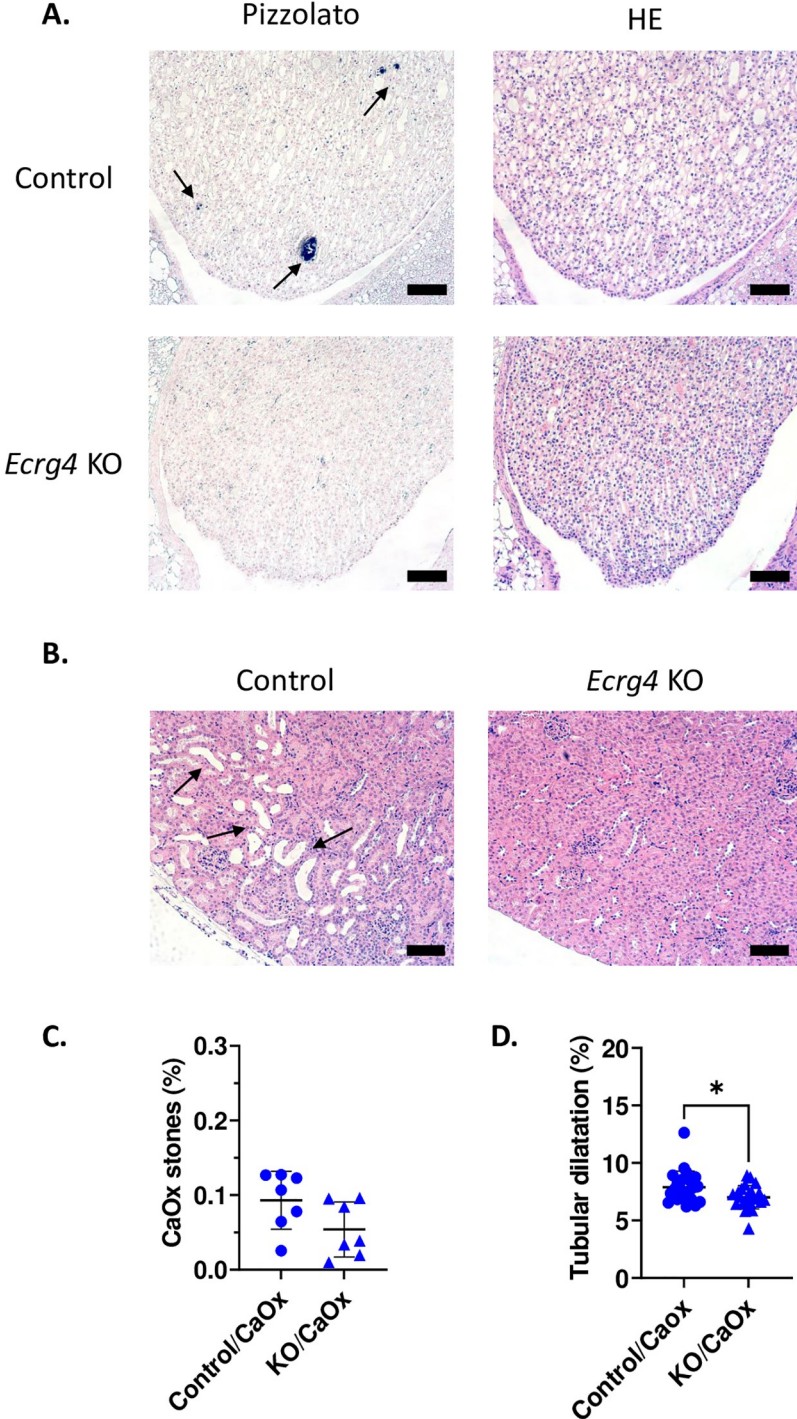

**Fig 4. *Ecrg4* KO male mice have less renal tubular obstruction after 8 days of CaOx diet. (A)** Representative pictures of renal papilla stained by Pizzolato (left panels) or Hematoxylin-eosin (HE, right panels) at 20x magnification. Intraluminal crystals and tubular dilation are visible in control mice but not in *Ecrg4* KO mice. Bars represent 100um. **(B)** HE staining of sections of the cortex of kidneys from control and *Ecrg4* male mice kidneys, after 8 days of CaOx diet, at 20x magnification. Bars represent 100um. The black arrows indicate the intraluminal crystals (A), as well as the tubular dilatation (B) in the control mice. **(C)** Quantification of the CaOx crystals in kidney tissue expressed as percentage of crystal surface over the kidney section (n = 7). **(D)** Quantification of tubular dilation of the cortex expressed as percentage of surface dilated over the whole surface (5 sections/mouse; n = 7). Data are shown individually and as mean ± SD. P values are calculated using t-test.

*Slc8A1*, *Aqp2*, *Pvalb*, *Casr*, *Vdr* was found between the kidneys of *Ecrg4* KO and control mice exposed to the CaOx diet (S13 Fig). However, a marked increase of PTHR1 can be seen in the kidney tissue of *Ecrg4* KO mice compared to controls (S14 Fig). In addition, genes involved in *Ecrg4*-dependent downstream pathways (*Nfkb1*, *Nfkb2*, *P53* and *Cox2*) were unchanged, with the notable exception of *P53*, *which* was less expressed in the KO mice (S15A Fig).

Finally, we exposed mouse cortical collecting duct cells (mCCD) to different calcium oxalate crystal concentrations (0, 300 or 600 ug of crystal/ml) and *Ecrg4* mRNA level was measured by qPCR (S15B Fig). We observed a dose-dependent decrease of *Ecrg4* mRNA upon treatment with higher crystal concentrations, as well as a decrease in *Nfkb1* expression when cells were exposed to the highest calcium oxalate crystal concentration (S15C Fig). No other change in the expression levels of the *Ecrg4* downstream genes (*Nfkb1*, *Nfkb2*, *P53* and *Cox2*) was observed.

## Discussion

This study investigated the expression of *Ecrg4* in the renal tissue, its regulation and its potential role in the development of kidney stones. We found expression of *Ecrg4* along the distal tubules, showed upregulation of *Ecrg4* mRNA expression in several chronic hypercalciuric mouse models and showed less signs of tubular obstruction in *Ecrg4* KO mice fed a calcium-oxalate rich diet. This promotes *Ecrg4* as a modulator of renal crystallization.

### *Ecrg4's* expression in renal tissue

We detected the presence of *Ecrg4* mRNA in the distal segments of the nephron (TAL, DCT/CNT and CD), while the proximal tubule was negative for *Ecrg4's* expression, in line with a recently published mouse single-cell RNA-seq analysis [13].

We spent substantial energy to detect and quantify ECRG4's protein expression in the kidney and other organs (S2–S7 Figs), but were unsuccessful to detect any specific signal. This could be explained by the fact that the specificity of the commercial antibodies used in this study has been tested only *a minima* by the providing companies, usually only in cell lines overexpressing ECRG4, and were not validated in native tissues obtained from control and *Ecrg4* KO mice. An alternative explanation for the lack of detection might be that ECRG4 is processed very quickly *in vivo* or that posttranslational modifications might prevent its detection. Indeed, the 148 amino acids full-length protein was shown to be cleaved into at least eight peptides of varying molecular weight, ranging from ~ 17 kDa to 2 kDa [14]. This process might render ECRG4's detection difficult with antibodies developed against full-length ECRG4. We thus looked for alternative methods that could help us identify the ECRG4 protein. A mass spectrometry analysis was performed on various protein extracts, from heart, brain, and *Ecrg4*-transfected HEK cells (S1 and S2 Tables). However, the detection level of ECRG4 fragments was discouragingly low, even in the overexpressing HEK cell system. We thus concluded that the ECRG4 protein expression is too low to be identified by the available tools or that only fragments of ECRG4 should be looked for. We thus continued our study only by interpreting the mRNA level of expression of *Ecrg4*.

### *Ecrg4* is regulated in models of hypercalciuria

We have previously identified *Ecrg4* as an mRNA upregulated in the hypercalciuric kidney-specific *Ncx1* KO mouse model. Here, we confirmed that *Ecrg4* expression is higher in the kidney of *Ncx1* KO mice compared to controls in all tubular segments expressing *Ecrg4*. We looked whether this regulation was observed in other models of hypercalciuria. We used the hypercalciuric *Cldn2* KO mouse model [11], lacking proximal tubule calcium reabsorption,

and mice fed the vitamin D analog, DHTS, which causes hypercalcemia and hypercalciuria. In these two models, up to 2.5 times increase in renal *Ecrg4* mRNA expression was observed compared to controls. By contrast, no change in the *Ecrg4* level was found when mice were injected acutely or for six consecutive days with furosemide, a known inducer of calciuria. We speculate that this discrepancy might be due to the relative short term and transient hypercalciuria induced by furosemide compared to chronic and sustained hypercalciuria of the other models. Thus, we propose that *Ecrg4's* increased expression is triggered only by chronic and sustained hypercalciuria.

The mechanism by which *Ecrg4* is regulated by urine calcium remained undetermined. We explored whether the calcitropic hormones, PTH or $1,25(OH)_2$-vitamin D, might have a direct effect on *Ecrg4*'s expression. We did not observe any change in *Ecrg4* renal expression after short term exposure to the two hormones. This might be due to the relative short stimulation time (2h for PTH and 6h for $1,25(OH)_2$-vitamin D), even if *Cyp24a1* and *Cyp27b1* transcript expression, used as controls, showed substantial increase after exposure to either of these hormones [9]. As previously noted, long-term exposure to the vitamin D analog, DHTS, induced a strong *Ecrg4* upregulation.

Based on this piece of data, we conclude that *Ecrg4* is regulated by long term exposure to sustained hypercalciuria, with no direct short-term effect of furosemide, PTH or $1,25(OH)_2$-vitamin D. The precise mechanism by which tubular calcium regulates *Ecrg4* expression remains to be identified.

## Loss of *Ecrg4* protects against tubular calcifications in male mice

We used a diet rich in calcium and in the oxalate precursor, hydroxyproline, for 8 days to induce hypercalciuria, hyperoxaluria, and calcium-oxalate nephropathy. In agreement with previous studies [15], we showed that kidneys of male mice developed a CaOx crystal nephropathy, while females were resistant, despite similar high urinary calcium excretion. Of note, sex differences in stone formation have been observed in humans and were attributed to sex hormones [16–20]. These differences seem to be more pronounced in mice and will need further studies to be deciphered. On the same note, our results show a gender-specific difference in *Ecrg4*'s expression in the renal tissue, under control conditions (S1 Fig). Consistent with previously published evidences, increased *Ecrg4* expression inhibits cell proliferation [21], which was found to account for renal crystal deposits in the setting of supersaturated urines [22, 23]. Further studies should assess if the difference in *Ecrg4* levels is directly involved in the different outcome in calcium-oxalate nephropathy between male and female mice.

To our surprise, we found that male mice devoid of *Ecrg4* and exposed to the CaOx diet had less tubular dilation and intra-renal calcifications. In these mice, we also found a preserved renal function, indicating that overall the mice were protected from tubular obstruction by calcium oxalate crystals. This protection could be due to the role of *Ecrg4* in the kidney or in other organs. The whole body constitutive knockout mouse model we used did not allow us to discriminate between the two possibilities. However, some observations suggest that the loss of *Ecrg4* may play a protective role by both renal and extra-renal effects. Hence, when exposed to the CaOx diet, control male mice had increased calcemia and calciuria compared to *Ecrg4* KO male mice. This suggests that *Ecrg4* KO mice adapt to the CaOx challenge by controlling their calcemia and calciuria by an unknown mechanism, potentially by decreasing their intestinal calcium absorption. Moreover, *Ecrg4* KO mice displayed lower fractional excretion of calcium, indicative of a direct effect of ECRG4 on the tubular calcium reabsorption. In an attempt to understand the mechanisms by which *Ecrg4* KO mice might have a different renal calcium handling, we analyzed the mRNA expression (S13 Fig), as well as protein level (S14 Fig) of

several renal genes and proteins involved in calcium homeostasis. We found similar levels between KO and control mice for all the tested candidates, except for the PTHR1 which was significantly increased in the *Ecrg4* KO mice. In these mice, this might indicate a possible adaptive mechanism to regulate calcium homeostasis that involves PTH/PTHR1 signaling pathway in the distal segment of the nephron.

Altogether, systemic as well as renal mechanisms could concur to alleviate crystal formation in *Ecrg4* KO mice under enriched calcium and hydroxyproline diet.

As *Ecrg4* is a tumor suppressor gene involved in the p53 and NFKB pathways [24–26], we wondered whether *Ecrg4* might be involved in mediating inflammation or tubular cell renewal, explaining the attenuated damages seen in the kidneys of *Ecrg4* KO mice. First, we found decreased expression of *p53* mRNA in the kidneys of *Ecrg4* KO mice (S15A Fig). This suggests that under the setting of CaOx diet challenge, male mice might have more cellular proliferative capacity to cope with crystal-induced injuries. Second, we found a down-regulation of the *Nfkb1* gene in the mouse cortical collecting duct cell line (mCCD cells) exposed to calcium-oxalate monohydrate crystals (S15B and S15C Fig). In this system, the decrease of *Nfkb1* expression might indicate that crystal-induced inflammation is reduced in the absence of *Ecrg4*. Indeed, it was previously shown that ECRG4 peptide induces the expression of proin-flammatory factors via a variety of intracellular signaling pathways. For example, increased reactive oxygen species levels, activated NFkB signaling pathway and interaction with the innate immunity receptor complex TLR4/CD14/MD2 complex, was demonstrated upon ECRG4 binding to LOX-1, and other scavenger receptors, such as Sccarf1, Cd36 and Stabili-1 [27, 28]. Altogether, deletion of *Ecrg4* might limit the defense mechanisms against crystal-induced injuries. However, how cells sense hypercalciuria and increase *Ecrg4* expression still remains to be identified.

## Strengths and limitations

We are not aware of any other study addressing the role of *Ecrg4* in the kidney and showing that *Ecrg4* is involved in renal calcium handling and in the risk of calcification. However, several limitations warrant mention. First, our evidence is mainly based on mRNA expression levels. As discussed previously, we were not able to detect ECRG4 full-length protein or fragments with the available anti-ECRG4 antibodies. Second, the mechanisms by which loss of *Ecrg4* reduces tubular calcifications remain open. We observed lower calciuria in male mice under CaOx diet compared to controls, but whether this might be sufficient to decrease intra-tubular crystal formation or retention remains unsolved. Finally, further studies will need to examine whether the limited renal insufficiency and hypercalciuria in the *Ecrg4* KO mice is rather time dependent, or if there is a long-term adaptation mechanism, which might involve some other organs than the kidney.

By showing that *Ecrg4* is involved in urine calcium handling and crystal nephropathy, we hope to open new perspectives of research in the field.

## Supporting information

**S1 Fig. *Ecrg4* mRNA expression is higher in female tissues compared to male tissues.** *Ecrg4* mRNA expression in the indicated organs relative to *Actb* (n = 2–3). Data are shown as individual dots and mean ± SD. Stars indicate *p<0.05, calculated using Student t-test.
(TIF)

**S2 Fig. Antibodies tested on total brain extract.** The equivalent of 50ug of protein was loaded on 13% SDS gel. The LS-Bio antibody (diluted 1:500, #LS-C172856), the Phoenix antibody

(diluted 1:300, #012–25), the Sigma antibody (diluted 1:500, #HPA008546) and the Santa Cruz antibody (diluted 1:500, # H-118) were used. Actin (diluted 1:500, #A2066) was used as loading control.
(TIF)

**S3 Fig. Antibodies tested on total heart extract.** The equivalent of 50ug of protein was loaded on 13% SDS gel. The LS-Bio antibody (diluted 1:500, #LS-C172856), the Phoenix antibody (diluted 1:300, #012–25), the Sigma antibody (diluted 1:500, #HPA008546) and the Santa Cruz antibody (diluted 1:500, # H-118) were used. Actin (diluted 1:500, #A2066) was used as loading control.
(TIF)

**S4 Fig. Antibodies tested on total esophagus extract.** 100ug of protein was loaded on 13% SDS gel. The LS-Bio antibody (diluted 1:500, #LS-C172856), the Phoenix antibody (diluted 1:300, #012–25), the Sigma antibody (diluted 1:500, #HPA008546) and the Santa Cruz antibody (diluted 1:500, # H-118) were used. Actin (diluted 1:500, #A2066) was used as loading control.
(TIF)

**S5 Fig. Antibodies tested on total kidney extract.** 30ug of protein was loaded on 13% SDS gel. The LS-Bio antibody (diluted 1:500, #LS-C172856), and the Santa Cruz antibody (diluted 1:500, # H-118) were used. Actin (diluted 1:500, #A2066) was used as loading control.
(TIF)

**S6 Fig. Immunofluorescence staining on kidney tissue using different antibodies.** Scale bars represent 200um. The LS-Bio antibody (diluted 1:100, #LS-C172856), the Sigma antibody (diluted 1:100, #HPA008546) and the Santa Cruz antibody (diluted 1:100, # H-118) were used.
(TIF)

**S7 Fig. Immunohistochemistry staining on different organ's tissue.** The Sigma antibody (diluted 1:50, #HPA008546) was used.
(TIF)

**S8 Fig. Calcium excretion after acute furosemide injection.** Mice were injected with 20mg/kg body weight furosemide or vehicle, and sacrificed after 15, 30, 60 and 240 min. P values are calculated using one-way ANOVA test, with Dunnett correction for multiple comparison. Each time point was compared to the vehicle. Data are shown as mean ± SD. Stars indicate * p<0.05.
(TIF)

**S9 Fig. Female *Ecrg4* KO mouse model characterization. (A)** Increase in the urinary calcium-creatinine ratio in both control and *Ecrg4* KO female mice upon CaOx diet. The ratios were calculated from the urine collected at baseline, and after 1 day and 8 days of exposure to either chow diet or the CaOx diet. **(B)** Urinary sodium-creatinine ratio in both control and *Ecrg4* KO female mice. **(C)** Urinary calcium/sodium excretion at day 8 of the CaOx diet. Data are shown as mean ± SD. p value was calculated using two-way ANOVA with Tukey correction for multiple comparison. Stars indicate * p<0.05 between the control and *Ecrg4* KO, § indicates p<0.05 between the controls of different diets, # indicates p<0.05 between the *Ecrg4* KO of different diets.
(TIF)

**S10 Fig. Mouse parameters during metabolic cages. (A)** Body weight, **(B)** food and **(C)** water intake parameters were measured over 24h period for males and females. At baseline, all mice were fed the chow diet and then exposed to either CaOx diet or chow diet (n = 4–5). Data

are shown as mean ± SD. p value was calculated using two-way ANOVA with Tukey correction for multiple comparison. Stars indicate * p<0.05 between the control and *Ecrg4* KO, § indicates p<0.05 between the controls of different diets, # indicates p<0.05 between the *Ecrg4* KO of different diets.
(TIF)

**S11 Fig. 24h urinary parameters.** Urinary phosphate **(A)**, potassium **(B)** excretion, osmolality **(C)** and pH **(D)**. Mice were individually housed in metabolic cages, after exposure to chow or CaOx diet (n = 4–5). Data are shown as mean ± SD. p value was calculated using two-way ANOVA with Tukey correction for multiple comparison. Stars indicate * p<0.05 between the control and *Ecrg4* KO, § indicates p<0.05 between the controls of different diets, # indicates p<0.05 between the *Ecrg4* KO of different diets.
(TIF)

**S12 Fig. No calcium oxalate crystals in the kidney of female mice.** Representative pictures of the Pizzolato staining of the **(A)** control and **(B)** *Ecrg4* KO female mice kidneys, under the CaOx. No calcium oxalate crystals can be observed in either of the kidney samples. Scale bars represent 1mm.
(TIF)

**S13 Fig. Genes involved in calcium handling are unchanged between control and *Ecrg4* KO male mice.** Relative expression of several renal genes involved calcium handling following the CaOx diet in *Ecrg4* KO mice is illustrated relative to *Actb* and to controls (1, dashed line) (n = 4). Data are shown as mean ± SD.
(TIF)

**S14 Fig. Relative quantification of selected proteins from kidney extracts.** Western blot analysis of (A) CLDN2, TRPV5, PVALB, CASR proteins in male mice kidney tissue (n = 3). Mice were fed either the chow diet or challenged with the CaOx diet. Western blot analysis of (B) PTHR1 protein in control vs *Ecrg4* KO male mice kidney tissue (n = 4–5). Mice were challenged with the CaOx diet. Quantities represented by the gel bands are expressed as intensity relative to β-Actin. Quantities represented by the gel bands are expressed as intensity relative to β-actin. All relative intensity results are presented as the means ± SD. Stars indicates * p<0.05. P values are calculated using two-way ANOVA test, with Sidak's correction for multiple comparison (A) or Student t-test (B).
(TIF)

**S15 Fig. mCCD cells stimulated with CaOx monohydrate crystals. (A)** The downstream genes that are activated upon ECRG4 signaling were quantified by qPCR in kidney extracts from control and *Ecrg4* KO male mice. Stars indicates * p<0.05. P values are calculated using Student t-test (n = 4). **(B)** mCCD cells were incubated with CaOx monohydrate crystals for 16h. The *Ecrg4* mRNA expression is shown as response to different crystals concentrations. **(C)** The downstream genes that are activated upon ECRG4 signaling were quantified in the mCCD cells upon calcium-oxalate monohydrate crystals incubation. Stars indicates * p<0.05. P values are calculated using one-way ANOVA comparing the mean of each CaOx crystal dosage (300 and 600 ug/ml CaOx) with the mean of the control column at 0 ug/ml CaOx crystal dosage (n = 3). Data are shown as mean ± SD.
(TIF)

**S1 Table. Identified proteins by mass spectrometry in control heart and brain tissues.**
(DOCX)

**S2 Table. Identified proteins by mass spectrometry in transfected HEK cells.**
(DOCX)

**S1 File.**
(DOCX)

**S1 Raw images.**
(PDF)

**S2 Raw images.**
(PDF)

**S3 Raw images.**
(PDF)

**S4 Raw images.**
(PDF)

## Acknowledgments

The authors thank Dr. Manfredo Quadroni (The Lausanne Protein Analysis Facility) for help with the mass spectrometry analysis and interpretation, as well as for helpful discussions.

The excellent work by Jean-Christophe Stehle, from the Animal Pathology Platform is acknowledged.

## Author Contributions

**Conceptualization:** Daniela Cabuzu, Suresh K. Ramakrishnan, Olivier Bonny.

**Data curation:** Daniela Cabuzu.

**Formal analysis:** Daniela Cabuzu, Olivier Bonny.

**Funding acquisition:** Olivier Bonny.

**Methodology:** Daniela Cabuzu, Olivier Bonny.

**Project administration:** Olivier Bonny.

**Resources:** Olivier Bonny.

**Supervision:** Suresh K. Ramakrishnan, Olivier Bonny.

**Validation:** Suresh K. Ramakrishnan, Olivier Bonny.

**Visualization:** Daniela Cabuzu, Olivier Bonny.

**Writing – original draft:** Daniela Cabuzu, Olivier Bonny.

**Writing – review & editing:** Daniela Cabuzu, Matthias B. Moor, Dusan Harmacek, Muriel Auberson, Fanny Durussel, Olivier Bonny.

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
