## [Decision Letter · Decision Letter 0]

30 May 2022

PONE-D-22-06861Loss of Ecrg4 improves calcium oxalate nephropathyPLOS ONE

Dear Dr. Bonny,

Thank you for submitting your manuscript to PLOS ONE. After careful consideration, we feel that it has merit but does not fully meet PLOS ONE’s publication criteria as it currently stands. Therefore, we invite you to submit a revised version of the manuscript that addresses the points raised during the review process.Please address the questions raised be the two reviewers.

We look forward to receiving your revised manuscript.

Kind regards,

Franziska Theilig

Academic Editor

PLOS ONE

Journal Requirements:

Reviewers' comments:

Reviewer's Responses to Questions

**Comments to the Author**

1. Is the manuscript technically sound, and do the data support the conclusions?

Reviewer #1: Yes

Reviewer #2: Partly

2. Has the statistical analysis been performed appropriately and rigorously? 

Reviewer #1: Yes

Reviewer #2: Yes

3. Have the authors made all data underlying the findings in their manuscript fully available?

Reviewer #1: Yes

Reviewer #2: Yes

4. Is the manuscript presented in an intelligible fashion and written in standard English?

Reviewer #1: Yes

Reviewer #2: Yes

5. Review Comments to the Author

Reviewer #1: 1.The paper may open a new way to explore the etiology of calcium oxalate nephropathy and maybe provide a new therapeutic strategy in Kidney stones.Fluent writing and correct grammar.The manuscript presented in an intelligible fashion and written in standard English.

2.The most regret is that we cannot detect ECRG4 protein expression in kindey.ECRG4 is a secreted protein that undergoes cleavage after secretion.The protein is specifically expressed in a manner dependent on differentiation status.Some paper found that Anti-ECRG4 immunohistochemical staining of rat genitourinary system, some tubular and glomerular epithelial cells show faint or moderate immunostaining(Eur J Histochem. 2015 May 18;59(2):2458;Gene. 2009 Dec 1;448(1):7-15).How to explain these differences?

3.Fig1-3 should indicate the number of samples in ordinate,Fig4 should indicate the magnification,and mark where is the crystal in Fig4A

4.in 358 line, in male mice not in male

5.Is there gender differences in the expression of ECRG4 in mice , except for hormonal in the part of discusson in 362 line?

6.How is upregulation of ECRG4 gene?It maybe better if we had experiments that upregulated Ecrg4 genes .

Reviewer #2: This study characterized the expression of Ecrg4 mRNA in different tissues as well as the expression pattern along the renal segment. The manuscript is very well written. The experimental approach is straightforward and clearly described. By in vivo and in vitro studies, authors reported the potential role of Ecrg4 in calcium oxalate nephropathy. Although there are several interesting concepts put forward by the data, there remains several unanswered questions and the underlying mechanisms are elusive. For this reason, I have some suggestions to improve the manuscript:

Major:

1. In response to hypercalciuria, Ecrg4 mRNA is upregulated, however this is not seen in furosemide-induced calciuria. In addition, renal Ecrg4 mRNA is not changed by PTH or 1,25(OH)3-VD. Therefore, it is speculated that beside directly sensing urine Ca concentration, renal Ecrg4 mRNA can also respond to other unknown local or systemic factors. Discussion about the underlying mechanisms for different hypercalciuric models and the resulting possible systemic changes could help to explain the discrepancy in renal Ecrg4 mRNA response to various hypercalciuric models. In addition, have author measured the Ecrg4 mRNA expression in control mice on calcium-oxalate diet, which induces hypercalciuria.

2. Is there sex difference in Ecrg4 mRNA expression, given that female mice are resistant to crystal formation in all conditions?

3. Plasma Ca concentration is comparable between the genotypes on calcium-oxalate diet, Ecrg4 KO shows lower urinary Ca excretion, which might indicate higher renal Ca reabsorption, meanwhile, mice might have lower intestinal Ca absorption to maintain the plasma Ca homeostasis. The authors measured mRNA expression of genes related to renal calcium handling, and found no differences between genotypes. mRNA expression is not always consistent with the protein abundance, therefore, WB to check the protein levels should provide more precise information. In addition, have the authors checked the hormone levels related to Ca homeostasis, PTH, VD3, FGF23, in the plasma of KO mice on calcium-oxalate diet compared to control mice?

4. in the line 358, the subhead “Ecrg4 protects against tubular calcification in males”. Here the conclusion is confusing. It is Ecrg4 KO mice show lower tubular dilation and nephrocalcinosis, means that loss of Ecrg4 has protective effect but not Ecrg4 itself. Same problem occurs in the later discussion section, in line 397 “Second, the mechanism by which Ecrg4 reduces tubular calcifications remains open”. Loss of Ecrg4 reduces injury rather Ecrg4 itself; In line 385-386, “In this system, the decrease of Nfkb1 expression might indicate that crystal-induced inflammation is reduced in presence of Ecrg4”, based on current results that decrease in Nfkb1 expression is in parallel with reduced Ecrg4 mRNA expression, so precisely, the possibly reduced inflammation (no evidence here) is due to decreased Ecrg4 mRNA expression rather “in presence of Ecrg4”. In contrast, the absence of Ecrg4 might have anti-inflammatory effect based on authors’ conclusions in this study, which is consistent with the following discussion that ECRG4 peptide is proinflammatory.

Minor

5. Y axis legend of S13A Fig is wrong, please correct it. It is not Ecrg4 expression.

6. In result section, line 313-314, “without major changes in the expression levels of 314 the ECRG4 downstream genes Nfkb1, Nfkb2, P53 and Cox2”. However, Nfkb1 mRNA expression is significantly reduced on 600 ug crystal and author should make it clear and clarify it because authors used this result in later discussion.

6. PLOS authors have the option to publish the peer review history of their article (what does this mean?). If published, this will include your full peer review and any attached files.

Reviewer #1: No

Reviewer #2: **Yes: **Jianxiang Xue

---

## [Author Response · Author response to Decision Letter 0]

8 Aug 2022

Dear Editor,

Dear Reviewers,

We are pleased to provide a revised version of our manuscript based on the constructive comments received. We think that the manuscript improved substantially now and we are thankful for the suggestions.

We performed additional experiments as suggested: we compared the expression levels of Ecrg4 mRNA in male and female tissues and we measured the protein level of calcium transport proteins by WB. This strengthened our conclusions.

We hope the manuscript is now in a shape allowing its publication.

Please find hereafter the detailed responses to the Reviewers’comments.

Best regards

O. Bonny

Reviewer #1: 

1. The paper may open a new way to explore the etiology of calcium oxalate nephropathy and maybe provide a new therapeutic strategy in Kidney stones. Fluent writing and correct grammar. The manuscript presented in an intelligible fashion and written in standard English.

Answer: 

Many thanks for your appreciation and positive comment. 

2. The most regret is that we cannot detect ECRG4 protein expression in kidney. ECRG4 is a secreted protein that undergoes cleavage after secretion. The protein is specifically expressed in a manner dependent on differentiation status. Some paper found that Anti-ECRG4 immunohistochemical staining of rat genitourinary system, some tubular and glomerular epithelial cells show faint or moderate immunostaining (Eur J Histochem. 2015 May 18;59(2):2458;Gene. 2009 Dec 1;448(1):7-15). How to explain these differences?

Answer: 

This was indeed a strong concern during the whole study and we tried to address this issue by different ways. We have actually followed the protocol published in the Eur J Histochem. 2015 May 18;59(2):2458 on different rat tissues. However, using the Ecrg4 KO mouse model we could not see any specific immunostaining on esophagus and kidney tissues. We infer that the antibody may not cross-react between mouse and rat.

Regarding the other publication (Gene 2009 Dec 1;448(1):7-15), the authors used 293T cells transfected with lentiviruses expressing myc-tagged ECRG4. An anti-myc antibody was used to detect the ECRG4 protein. No native ECRG4 protein was detected with specific antibody in this study.

Therefore, these differences can be explained by the different animal model/system used and by the lack of negative control to validate the antibodies used.

3. Fig1-3 should indicate the number of samples in ordinate, Fig4 should indicate the magnification, and mark where is the crystal in Fig4A

Answer: 

We changed the figure and legends according to the reviewer`s suggestion. 

4. in 358 line, in male mice not in male

Answer: 

We changed the text according to the reviewer`s comment. 

5. Is there gender differences in the expression of ECRG4 in mice, except for hormonal in the part of discussion in 362 line? 

Answer: 

We performed this experiment as suggested by this reviewer. We found that female mice have 2 times more Ecrg4 mRNA expression in the kidney and heart tissues compared to male mice.

We now introduced this data in the manuscript (Supplemental material, S1 Fig) and discussed it. 

Here is the modified text in the Discussion section:

``Of note, sex differences in stone formation have been observed in humans and were attributed to sex hormones (1-5). These differences seem to be more pronounced in mice and will need further studies to be deciphered. On the same line, our results show a gender-specific difference in Ecrg4`s expression in the renal tissue, under control conditions (S1 Fig). Further studies are needed to assess if the sex-difference in Ecrg4 expression level might be directly involved in the different outcome in calcium-oxalate nephropathy observed between male and female mice.``

6. How is upregulation of ECRG4 gene? It maybe better if we had experiments that upregulated Ecrg4 genes.

Answer: 

The precise mechanism by which tubular calcium regulates Ecrg4 expression remains to be identified. Based on our findings, we conclude that Ecrg4 is regulated only by long term exposure to sustained hypercalciuria, with no short-term effect of furosemide, PTH or 1,25(OH)2-vitamin D. 

Reviewer #2: 

This study characterized the expression of Ecrg4 mRNA in different tissues as well as the expression pattern along the renal segment. The manuscript is very well written. The experimental approach is straightforward and clearly described. By in vivo and in vitro studies, authors reported the potential role of Ecrg4 in calcium oxalate nephropathy. Although there are several interesting concepts put forward by the data, there remains several unanswered questions and the underlying mechanisms are elusive. For this reason, I have some suggestions to improve the manuscript:

Major:

1. In response to hypercalciuria, Ecrg4 mRNA is upregulated, however this is not seen in furosemide-induced calciuria. In addition, renal Ecrg4 mRNA is not changed by PTH or 1,25(OH)3-VD. Therefore, it is speculated that beside directly sensing urine Ca concentration, renal Ecrg4 mRNA can also respond to other unknown local or systemic factors. Discussion about the underlying mechanisms for different hypercalciuric models and the resulting possible systemic changes could help to explain the discrepancy in renal Ecrg4 mRNA response to various hypercalciuric models. In addition, have author measured the Ecrg4 mRNA expression in control mice on calcium-oxalate diet, which induces hypercalciuria.

Answer:

We showed that Ecrg4’s expression is upregulated in hypercalciuric mice when hypercalciuria is chronic and sustained (in the Ncx1 KO and Cldn2 KO mouse models or in mice under Vitamin D analog treatment), as opposed to transient hypercalciuria (furosemide treatment) or short-term exposure to calcitropic hormones (PTH, vitamin D injection). When quantifying the Ecrg4 mRNA expression in the kidney of mice fed the diet inducing hypercalciuria and hyperoxaluria compared to the chow diet, we observed a significant decrease in Ecrg4 expression. 

Given that hypercalciuria alone does not induce nephrocalcinosis, as in the case of the CaOx diet, we speculate that the difference in Ecrg4 expression could be explained by the differences in the mouse models. Indeed, we showed that the loss of Ecrg4 protects against calcium-oxalate nephropathy.

Moreover, there is published evidence that Ecrg4 may play a role in coordinating the inflammatory and proliferative cell response that could help maintain epithelium integrity, as seen by gene downregulation in traumatic brain injury rat model (6), as well as in human lung epithelial injury (7). 

Thus, the decreased Ecrg4 expression in the kidney of control mice challenged with the CaOx diet compared to chow diet may suggest a defense mechanisms against crystal-induced injuries.

2. Is there sex difference in Ecrg4 mRNA expression, given that female mice are resistant to crystal formation in all conditions? 

Answer: 

We performed this experiment and found that female mice have 2 times more Ecrg4 mRNA expression in the kidney and heart tissues compared to male mice.

We now introduced this data in the manuscript (figure in the Supplemental material, S1 Fig) and discussed it. 

Here is the modified text in the Discussion section:

``Of note, sex differences in stone formation have been observed in humans and were attributed to sex hormones (1-5). These differences seem to be more pronounced in mice and will need further studies to be deciphered. On the same line, our results show a gender-specific difference in Ecrg4`s expression in the renal tissue, under control conditions (S1 Fig). Further studies are needed to assess if the sex-difference in Ecrg4 expression level might be directly involved in the different outcome in calcium-oxalate nephropathy observed between male and female mice.``

3. Plasma Ca concentration is comparable between the genotypes on calcium-oxalate diet, Ecrg4 KO shows lower urinary Ca excretion, which might indicate higher renal Ca reabsorption, meanwhile, mice might have lower intestinal Ca absorption to maintain the plasma Ca homeostasis. The authors measured mRNA expression of genes related to renal calcium handling, and found no differences between genotypes. mRNA expression is not always consistent with the protein abundance, therefore, WB to check the protein levels should provide more precise information. In addition, have the authors checked the hormone levels related to Ca homeostasis, PTH, VD3, FGF23, in the plasma of KO mice on calcium-oxalate diet compared to control mice?

Answer: 

We thank the reviewer for this excellent suggestion. We are now providing the WB quantification for TRPV5, PVALB, PTHR1, CLND2 and CASR on kidney extract from control and Ecrg4 KO mice challenged with the CaOx diet. 

The Ecrg4 KO mice show lower urinary Ca excretion and comparable plasma Ca levels. While at the mRNA level we did not find any changes in renal calcium handling genes, at the protein level, we found a significant increase in PTHR1’s expression and a non-significant decrease in CASR between the control and Ecrg4 KO mice fed the CaOx diet. This might indicate a possible Ecrg4-mediated Ca reabsorption via the PTH/PTHR1signaling pathway in the distal parts of the nephron, with no impact on the renal reabsorption of Ca in the proximal tubule, given the CLDN2 unchanged levels. 

We introduced this data in the manuscript (Supplemental material, S14 Fig) and discussed it.

No hormones (PTH, VD3, FGF23) could be measured in the plasma of Ecrg4 KO mice and controls due to unavailability of blood samples during the revision period. 

4. In the line 358, the subhead “Ecrg4 protects against tubular calcification in males”. Here the conclusion is confusing. It is Ecrg4 KO mice show lower tubular dilation and nephrocalcinosis, means that loss of Ecrg4 has protective effect but not Ecrg4 itself. Same problem occurs in the later discussion section, in line 397 “Second, the mechanism by which Ecrg4 reduces tubular calcifications remains open”. Loss of Ecrg4 reduces injury rather Ecrg4 itself; In line 385-386, “In this system, the decrease of Nfkb1 expression might indicate that crystal-induced inflammation is reduced in presence of Ecrg4”, based on current results that decrease in Nfkb1 expression is in parallel with reduced Ecrg4 mRNA expression, so precisely, the possibly reduced inflammation (no evidence here) is due to decreased Ecrg4 mRNA expression rather “in presence of Ecrg4”. In contrast, the absence of Ecrg4 might have anti-inflammatory effect based on authors’ conclusions in this study, which is consistent with the following discussion that ECRG4 peptide is proinflammatory.

Answer: 

We thank the reviewer for pointing an important misunderstanding. We corrected the text accordingly in the Discussion. 

Minor

5. Y axis legend of S13A Fig is wrong, please correct it. It is not Ecrg4 expression.

Answer: 

We corrected it. Thank you for pointing this issue. 

6. In result section, line 313-314, “without major changes in the expression levels of 314 the ECRG4 downstream genes Nfkb1, Nfkb2, P53 and Cox2”. However, Nfkb1 mRNA expression is significantly reduced on 600 ug crystal and author should make it clear and clarify it because authors used this result in later discussion.

Answer: 

We corrected the text accordingly. 

Here is the modified text:

“Finally, we exposed mouse cortical collecting duct cells (mCCD) to different calcium oxalate crystal concentrations (0, 300 or 600 ug of crystal/ml) and Ecrg4 mRNA level was measured by qPCR. We observed a dose-dependent decrease of Ecrg4 mRNA upon treatment with higher crystal concentrations, as well as a decrease in Nfkb1 expression when cells were exposed to the highest calcium oxalate crystal concentration. No other changes in the expression levels of the ECRG4 downstream genes (Nfkb1, Nfkb2, P53 and Cox2) was observed (S15B and C Fig).”

References

1. Scales CD, Jr., Curtis LH, Norris RD, Springhart WP, Sur RL, Schulman KA, et al. Changing gender prevalence of stone disease. J Urol. 2007;177(3):979-82.

2. Schwille PO, Manoharan M, Schmiedl A. Is idiopathic recurrent calcium urolithiasis in males a cellular disease? Laboratory findings in plasma, urine and erythrocytes, emphasizing the absence and presence of stones, oxidative and mineral metabolism: an observational study. Clin Chem Lab Med. 2005;43(6):590-600.

3. Fan J, Chandhoke PS, Grampsas SA. Role of sex hormones in experimental calcium oxalate nephrolithiasis. J Am Soc Nephrol. 1999;10 Suppl 14:S376-80.

4. Yagisawa T, Ito F, Osaka Y, Amano H, Kobayashi C, Toma H. The influence of sex hormones on renal osteopontin expression and urinary constituents in experimental urolithiasis. J Urol. 2001;166(3):1078-82.

5. Ko B, Bergsland K, Gillen DL, Evan AP, Clark DL, Baylock J, et al. Sex differences in proximal and distal nephron function contribute to the mechanism of idiopathic hypercalcuria in calcium stone formers. American Journal of Physiology-Regulatory, Integrative and Comparative Physiology. 2015;309(1):R85-R92.

6. Podvin S, Roberton A, Johanson C, Stopa E, Eliceiri B, Baird A. Augrin, Ecilin and Argilin: Characterization of neuropeptide candidates encoded by the esophageal cancer related Gene-4 (ecrg4) and their localization in the mouse choroid plexus. JSfNP 2009;85.

7. Kao S, Shaterian A, Cauvi DM, Dang X, Chun HB, De Maio A, et al. Pulmonary preconditioning, injury, and inflammation modulate expression of the candidate tumor suppressor gene ECRG4 in lung. Experimental Lung Research. 2015;41(3):162-72.

---

## [Decision Letter · Decision Letter 1]

27 Sep 2022

Loss of Ecrg4 improves calcium oxalate nephropathy

PONE-D-22-06861R1

Dear Dr. Bonny,

We’re pleased to inform you that your manuscript has been judged scientifically suitable for publication and will be formally accepted for publication once it meets all outstanding technical requirements.

Kind regards,

Franziska Theilig

Academic Editor

PLOS ONE

Additional Editor Comments (optional):

Reviewers' comments:

Reviewer's Responses to Questions

**Comments to the Author**

1. If the authors have adequately addressed your comments raised in a previous round of review and you feel that this manuscript is now acceptable for publication, you may indicate that here to bypass the “Comments to the Author” section, enter your conflict of interest statement in the “Confidential to Editor” section, and submit your "Accept" recommendation.

Reviewer #1: All comments have been addressed

Reviewer #2: All comments have been addressed

2. Is the manuscript technically sound, and do the data support the conclusions?

Reviewer #1: Yes

Reviewer #2: Yes

3. Has the statistical analysis been performed appropriately and rigorously? 

Reviewer #1: Yes

Reviewer #2: Yes

4. Have the authors made all data underlying the findings in their manuscript fully available?

Reviewer #1: Yes

Reviewer #2: Yes

5. Is the manuscript presented in an intelligible fashion and written in standard English?

Reviewer #1: Yes

Reviewer #2: Yes

6. Review Comments to the Author

Reviewer #1: Thw authors had already addressed my comments raised in a previous round of review and revised the manuscript .I recommend to accept the manuscript and had no further suggestion for the authors .

Reviewer #2: The authors fully addressed the raised comments by revision, therefore, I have no more comments or concerns on the manuscript.

7. PLOS authors have the option to publish the peer review history of their article (what does this mean?). If published, this will include your full peer review and any attached files.

Reviewer #1: No

Reviewer #2: No

---

## [Editor Report · Acceptance letter]

4 Oct 2022

PONE-D-22-06861R1 

Loss of *Ecrg4* improves calcium oxalate nephropathy 

Dear Dr. Bonny:

I'm pleased to inform you that your manuscript has been deemed suitable for publication in PLOS ONE. Congratulations! Your manuscript is now with our production department. 

Kind regards, 

on behalf of

Dr. Franziska Theilig 

Academic Editor

PLOS ONE